# iChip-Inspired Isolation, Bioactivities and Dereplication of *Actinomycetota* from Portuguese Beach Sediments

**DOI:** 10.3390/microorganisms10071471

**Published:** 2022-07-20

**Authors:** José Diogo Neves dos Santos, Susana Afonso João, Jesús Martín, Francisca Vicente, Fernando Reyes, Olga Maria Lage

**Affiliations:** 1Department of Biology, Faculty of Sciences, University of Porto, Rua do Campo Alegre S/N, 4169-007 Porto, Portugal; susanaaj15@gmail.com (S.A.J.); olga.lage@fc.up.pt (O.M.L.); 2Interdisciplinary Centre of Marine and Environmental Research, University of Porto, Terminal de Cruzeiros do Porto de Leixões, Avenida General Norton de Matos, S/N, 4450-208 Matosinhos, Portugal; 3Fundación MEDINA, Centro de Excelencia en Investigación de Medicamentos Innovadores en Andalucía, Avenida del Conocimiento, 34 Parque Tecnológico de Ciencias de la Salud, 18016 Granada, Spain; jesus.martin@medinaandalucia.es (J.M.); fvperez1958@gmail.com (F.V.); fernando.reyes@medinaandalucia.es (F.R.)

**Keywords:** iChip, antimicrobial activities, LC/HRMS, marine *Actinomycetota*

## Abstract

Oceans hold a stunning number of unique microorganisms, which remain unstudied by culture-dependent methods due to failures in establishing the right conditions for these organisms to grow. In this work, an isolation effort inspired by the iChip was performed using marine sediments from Memoria beach, Portugal. The isolates obtained were identified by 16S rRNA gene analysis, fingerprinted using BOX-PCR and ERIC-PCR, searched for the putative presence of secondary metabolism genes associated with polyketide synthase I (PKS-I) and non-ribosomal peptide synthetases (NRPS), screened for antimicrobial activity against *Escherichia coli* ATCC 25922 and *Staphylococcus aureus* ATCC 29213, and had bioactive extracts dereplicated by LC/HRMS. Of the 158 isolated strains, 96 were affiliated with the phylum *Actinomycetota*, PKS-I and NRPS genes were detected in 53 actinomycetotal strains, and 11 proved to be bioactive (10 against *E. coli*, 1 against *S. aureus* and 1 against both pathogens). Further bioactivities were explored using an “one strain many compounds” approach, with six strains showing continued bioactivity and one showing a novel one. Extract dereplication showed the presence of several known bioactive molecules and potential novel ones in the bioactive extracts. These results indicate the use of the bacteria isolated here as sources of new bioactive natural products.

## 1. Introduction

Oceans cover over approximately 70% of the Earth’s surface and are involved in every aspect of all biogeochemical cycles [1]. Moreover, it is estimated that the oceans contain around 6 gigatons of carbon [2] and are home to an incredible diversity of organisms. Yet, it is estimated that humans have only explored about 20% of the ocean [3]. Many of the biogeochemical processes that take place in the oceans are carried out by the present microbiological community. Bacteria, which are estimated to account for 10^6^ cells per millilitre of sea water [4], represent approximately 1.3 gigatons of carbon [2] and are thought to have a higher diversity than all the organisms existing in tropical rainforests [5]. As bacteria play major roles in maintaining the cycles of elements, bacteria have the potential to cause a revolution in all biotechnological fields, such as in the fight against global warming and even in the field of medicine [6].

Nature has long been the inspiration behind the search for novel medicines. Poultices, plant mixtures and tonics have been used since antiquity for treating multiple conditions. Evidence found on Nubian mummies points to the use of a mixture containing tetracycline in the possible treatment of fractures [7] and, most notably, extracts of willow bark (genus Salix) containing salicylic acid, a drug that was first isolated in 1828, were used as far back as the ancient Sumerians [8].

Marine bacterial natural products hold particular interest in fields such as antimicrobial and anticancer research. This can be ascribed to the fact that marine bacteria, namely members of *Actinomycetota*, *Pseudomonadota*, *Bacillota* and *Cyanobacteria,* are great reservoirs of bioactive molecules [9]. These include novel carbon skeletons, such as taromycin B [10] and janthinopolyenemycins [11], and molecules in clinical phase trials such as bryostatins [12] and salinisporamide A [13].

Many of these bioactive molecules are biosynthesised by two groups of enzymes, the polyketide synthases (PKS) and the non-ribosomal peptide synthetases (NRPS). PKSs are a family of enzyme complexes that produce polyketides similar to molecular LEGO, which are classified into three classes, types I, II and III [14]. On the other hand, NRPSs are a cluster of modular enzymes that catalyse peptides in a similar way as in ribosomal protein synthesis. Non-ribosomal peptide synthetases produce a peptide chain of two to forty-eight residues in length and are able to use a variety of standard and non-proteinogenic amino acids [15]. The presence of these genes in newly discovered organisms can indicate the possibility of discovering novel natural products.

This phylum is composed of Gram-positive, aerobic, nonmotile bacteria with a high G + C content in their DNA. *Actinomycetota* show a large range of morphologies, which can vary from spore-forming aerial mycelium to asporogenous rods and cocci. *Actinomycetota* can be found in most natural habitats, and the class *Actinomycetia*, and in particular, the *Actinomycetales*, are responsible for the production of over two-thirds of naturally derived antibiotics [16] and novel bioactive molecules isolated in the past 20 years [6,9]. In fact, salinisporamide A and taromycin B were isolated from bacteria belonging to *Actinomycetota*. Furthermore, *Actinomycetota* are a source of many enzymes responsible for a range of industrial and bioremediation processes [17]. These facts make this phylum predominant in biotechnological studies.

Metagenomic data have revealed that marine *Actinomycetota* display substantial diversity, including new subclasses and orders [18]. Initial theories postulated that marine *Actinomycetota* were, in fact, simply the run-off of spores and cells from terrestrial and freshwater sources [19]. However, the discovery of obligate marine taxa, such as *Salinispora* [20] and *Marinispora* [21], came to disprove this theory. Furthermore, many novel taxonomic strains are being isolated from diverse habitats that vary from mangroves [22] and kelp forests [23] to deep-sea sediments [24,25]. Moreover, genomic approaches have proven that deep-sea sediments may contain over a thousand different actinobacterial taxa, a significant percentage of which are believed to be novel taxonomic groups [26].

Conventional techniques for isolating *Actinomycetota* include physical and chemical treatments and nutritional formulations of the isolating media. Physical treatments can vary from the use of high temperatures (heat) or dryness, to selectively killing vegetative cells and stimulating the germination of spores, while chemical treatments with usual antibiotics are used to control the number of fast-growing bacteria [27]. Moreover, antifungal compounds such as cycloheximide are also used. Medium nutritional formulations are adjusted to include complex sugars and polymers, of which *Actinomycetota* are known biological degraders [28]. However, the domestication of new marine actinobacterial taxa can be challenging. Marine *Actinomycetota* have needed to adapt to the unique and extreme temperatures, pressure and nutritional conditions present in the ocean. This latter condition is particularly difficult to simulate under laboratory conditions, since micronutrients can differ significantly over time in nature as the microbiological community interacts with each other, meaning that many taxa remain to be brought into axenic culture [26,29]. Thus, new, smarter isolation techniques need to be applied to the isolation of novel bacteria. A promising technique is the isolation chip (iChip) [30]. The iChip allows for the in situ enrichment of cells, thus facilitating the domestication of new strains [31]. Briefly, this device comprises a plate with multiple wells, which are inoculated with a gelled suspension of environmental bacterial cells, covered by porous membranes and sealed, creating miniature diffusion chambers [30]. The cells are then incubated under their natural conditions, with particular regard to nutritional availability.

In this study, in situ conditions were simulated for an isolation effort inspired by the iChip technology. The isolation was performed using marine beach sediments from Memoria beach, on the northern coast of Portugal. The obtained isolates were identified based on 16S rRNA gene analysis. Furthermore, secondary metabolism genes associated with the enzymes PKS-I and NRPS were searched for and antimicrobial activity against *Escherichia coli* ATCC 25,922 and *Staphylococcus aureus* ATCC 29,213 tested in strains with relevant secondary metabolism genes.

## 2. Materials and Methods

### 2.1. Sampling and Isolation

Wet sediment samples were retrieved from Memoria beach in Perafita, Matosinhos (41°13′50.35″ N 8°43′16.61″ W), Portugal in March 2020. Approximately 25 g of sediments was collected in a sterile falcon tube and a recipient of 21 × 15 × 14 cm was about half filled with wet sediments from the same beach (Figure 1). The sediment samples were brought to the laboratory refrigerated. To simulate the miniature diffusion chambers present in the iChip culturing system [30], a MultiScreen^®^ 96-Well Filtration Plate was used. The MultiScreen^®^ 96-Well Filtration Plate has a 0.22 µm hydrophilic polyvinylidene fluoride filter on the bottom side.

Around 10 mL of sediment was added to a falcon tube and 10 mL of sterile seawater added to create a cell suspension. Cells were counted using a Thoma counting chamber, and a gelled suspension with 10 cells per 100 μL was prepared. This suspension was a mixture of sterile natural seawater with agar at 0.8% (*w*/*v*). One hundred microliters of this suspension were then seeded in each well of the 96-Well Filtration Plate. The plate’s upper lid was closed and sealed with Parafilm^®^ and the whole plate placed in a box, with the filter end of the plate covered with wet sediment from the beach. The box was kept in the dark, at room temperature. After incubation for 60 days, the bacterial growth in the gelled wells was inoculated into medium M600 (0.1% *w/v* peptone, 0.1% *w/v* yeast extract, 5 mM Tris-HCl pH 7.5, 0.1% *w/v* glucose, 0.1% *v/v* of vitamin solution (0.1 μg mL^−1^ cyanocobalamin, 2.0 μg mL^−1^ biotin, 5.0 μg mL^−1^ thiamine-HCl, 5.0 μg mL^−1^ Ca-pantothenate, 2.0 μg mL^−1^ folic acid, 5.0 μg mL^−1^ riboflavin, and 5.0 μg mL^−1^ nicotinamide) and 0.2% *v/v* of Hutner’s solution (99 mg/L FeSO_4_.7H_2_O, 12.67 mg/L NaMoO_4_.2H_2_O, 3.34 g/L CaCl_2_.2H_2_O, 29.70 g/L MgSO_4_.7H_2_O, 50 mL/L “44” metals solution, and 10.0 g/L nitrilotriacetic acid; for 100 mL of “44” metals: 250 mg ethylenediaminetetraacetic acid, 1095 mg ZnSO_4_.7H_2_O, 500 mg FeSO_4_.7H_2_O, 154 mg MnSO_4_.H_2_O, 39.2 mg CuSO_4_.5H_2_O, 24.8 mg Co(NO_3_)_2_.6H_2_O, and 17.7 mg Na_2_B4O_7_.10H_2_O) [32] and marine agar (MA) (0.5% *w/v* peptone, 0.1% *w/v* yeast extract, and 1L aged natural seawater) plates and incubated at 25 °C until colonies were observed and picked. When pure cultures were obtained, they were kept in the isolation medium (M600 or MA) and cryopreserved in M600 or marine broth supplemented with 20% (*v*/*v*) glycerol, at −80 °C.

### 2.2. Identification of the Strain’s Phylogeny and Detection of Secondary-Metabolite-Associated Genes

Pure cultures of the isolated strains were used for genomic DNA extraction. DNA was extracted with the E.Z.N.A.^®^ Bacterial DNA Isolation Kit (Omega Bio-Tek, Norcross, TN, USA) and strains were identified by analysing their 16S rRNA gene. This gene was amplified by polymerase chain reaction (PCR) with the primers 27F and 1492R, following the protocol described by Bondoso et al. [33]. PCR products were purified using the Illustra™ GFX™ PCR DNA and Gel Band Purification Kit and sequenced by Sanger sequencing at Eurofins Genomics. Sequences were analysed using Geneious R11 and the consensus 16S rRNA gene sequences obtained were deposited in the National Center for Biotechnology Information Search (NCBI) database. The phylogeny was inferred using the 16S-based ID tool in the EzBioCloud platform [34]. To further assist in the differentiation of actinobacterial strains, BOX-PCR and ERIC-PCR fingerprinting were performed on strains with the same 16S rRNA gene affiliation, using the primers BOX-A1R and ERIC1R and ERIC2, respectively. PCR protocols were performed as described in Lage et al. [35]. PCR products were separated by electrophoresis in a 1.5% agarose gel Tris–acetate–EDTA buffer, and phylogenetic dendrograms were constructed using PyElph with an unweighted pair group method with arithmetic mean for clustering the distance matrix [36].

For the detection of biosynthetic gene clusters, PCR amplification was performed with the degenerate primers MDPQQRf (5′-RTRGAYCCNCAGCAICG-3′) and HGTGTr (5′-VGTNCCNGTGCCRTG-3′) aimed at the α-keto synthase of PKS-I motif [37] and MTRF2 (5′-GCNGG(C/T)GG(C/T)GCNTA(C/T)GTNCC-3′) and DKF (5′-GTGCCGGTNCCRTGNGYYTC-3′) for core motif-V of NRPS [38]. PCR protocol was performed as described by Graça et al. [39]. The expected amplicon size of PKS-I and NRPS genes was 750 and 1000 bp, respectively.

### 2.3. Strain Fermentation and Extraction

The isolated *Actinomycetota* with amplified PKS-I, NRPS or both genes were chosen for antimicrobial screening. Cultures were fermented in plates containing 25 mL of modified M13 medium (M607) [32] for 15 days, at 25 °C, in the dark. The cultures were collected and steeped in 100 mL of ethyl-acetate overnight for bioactive molecule extraction. The suspension in ethyl-acetate was collected and the agar was washed twice with 10 mL ethyl-acetate, which was added to the extraction suspension. The ethyl-acetate was dried to completeness and the extract was dissolved in 500 µL of 20% (*v*/*v*) DMSO. Additionally, an unfermented medium contained in a Petri dish was extracted with the same protocol to serve as a medium control. To further explore the antimicrobial activity, bioactive strains were refermented in different media. These included modified M607 and M600, MA, CGY [40] and a 1:10 diluted version of M607 medium, and the cultures were extracted as described above.

### 2.4. Antimicrobial Screening

Antimicrobial screening of the extracts was performed against *E. coli* ATCC 25,922 and *S. aureus* ATCC 29,213 as described previously by Santos et al. [41]. Briefly, single colonies of each target microorganism were incubated in nutrient broth (NB) overnight, at 37 °C and 220 rpm. Cultures were then diluted to obtain an inoculum with 5.0 × 10^5^ cells/mL. Then, 90 µL/well of the corresponding diluted inoculum was mixed with 10 µL of extract in triplicate. Streptomycin and ampicillin at 10 mg/mL were used as positive controls for *E. coli* ATCC 25,922 and *S. aureus* ATCC 29213, respectively. Negative internal controls were included. These controls comprise solvent (DMSO) and bacterial growth controls and, additionally, medium controls. Absorbance (at 600 nm) was measured in a Thermo Scientific™ Multiskan™ GO. The percentage of growth inhibition was calculated using the following equation:(1)$$\%\mathrm{inhibition}=100-100\times \frac{\left(T_{\mathrm{FE}}-T_{0E}\right)-\left(T_{\mathrm{FB}}-T_{0B}\right)}{\left(T_{\mathrm{FG}}-T_{0G}\right)-\left(T_{\mathrm{FB}}-T_{0B}\right)}$$
where T_0_ is the absorbance at 0 h, T_F_ is the absorbance at 24 h, E is the extract well, B is blank wells and G is the control growth wells.

Each extract was tested three times on different days with new inocula (*n* = 3). Extracts were considered to have an inhibitory effect if the target growth was reduced by more than 50% in at least two assays and the average was also above the 50% threshold.

### 2.5. Dereplication of Extracts

Extract dereplication was performed by liquid chromatography/high-resolution mass spectroscopy (LC/HRMS) with an Agilent 1200 Rapid Resolution HPLC interfaced with a Bruker maXis mass spectrometer. The column used was a Zorbax SB-C8 column (2.1 mm × 30 mm, 3.5 mm particle size), with two solvents used for the mobile phase. Both solvents were composed of water and acetonitrile, in a 90:10 ratio for solvent A, and in a 10:90 ratio for B, and 13 mM ammonium formate and 0.01% trifluoracetic acid were added to both. The mass spectrometer was operated in positive ESI mode. To obtain putative component identification, the retention time and exact mass of the components were compared against Fundación MEDINA’s high-resolution mass spectrometry database. For the components with no matches in the MEDINA database, the predicted molecular formula and exact mass were searched for in the Chapman and Hall Dictionary of Natural Products (DNP) database. If a plausible match was found, considering the exact mass/molecular formula, the producing microorganism, and the target assay, the molecule was reported as a suggested component of the fraction [42].

## 3. Results

### 3.1. Isolation and Identification of Strains

Overall, 192 isolates were obtained, but only 158 were identified due to the lack of viability of several isolates. The 158 strains were identified through 16S rRNA gene-based analysis (Appendix A). In total, 79 strains were obtained in both M600 and MA. In strains isolated in M600, 45 belonged to *Actinomycetota*, 17 to *Pseudomonadota*, 12 to *Bacillota* and 4 to *Bacteroidota*. In the strains isolated in marine agar, 51 belonged to *Actinomycetota*, 16 to *Pseudomonadota*, 10 to *Bacillota* and 4 to *Bacteroidota*. All strains of 16S rRNA gene sequences were deposited in NCBI’s GenBank database with accession numbers MW739985 to MW740142.

In general, *Actinomycetota* made up the majority of the isolated strains (Figure 2 and Appendix A), in a total of 96 isolates corresponding to 60.8% of the sequenced isolates. Furthermore, 33 strains (20.9%) of the isolates belonged to *Pseudomonadota*, 22 strains (13.9%) to *Bacillota**,* and 7 strains (4.4%) to the *Bacteroidota* (Figure 2 and Appendix A). At the genus level (Appendix A), the majority belonged to the genus *Streptomyces* (48 isolates, 30%), followed by *Nocardiopsis* (29 isolates, 18.1%), *Bacillus* (19 isolates, 12.0%) and *Pseudoalteromonas* (9 isolates, 5.7%). Moreover, at least two or more strains were isolated from genera such as *Psychrobacter* (6 isolates, 3.75%), *Rhodococcus* (5 isolates, 3.125%), *Arthrobacter* (4 isolates, 2.5%), *Microbacterium*, *Aquimarina*, *Cobetia*, *Limimaricola*, *Sulfitobacter*, *Tritonibacter* (each with 3 isolates, 1.9%), and *Dietzia*, *Arenibacter*, *Alkalihalobacillus* and *Marinobacter* (each with 2 isolates, 1.3%). From the genera *Corynebacterium*, *Kocuria*, *Micromonospora*, *Nocardia*, *Plantibacter*, *Catalinimonas*, *Tenacibaculum*, *Fictibacillus*, *Henriciella*, *Phaeobacter*, *Providencia* and *Vibrio*, only one isolate (0.63% each) was obtained (Appendix A). Only one strain, PMIC_1C1B, showed a similar percentage below the species threshold of 98.70% [43], indicating that it possibly constitutes a novel taxon. When considering the media used for the isolation, 81 viable strains were obtained in medium M600, while in medium MA, 79 strains were retrieved (Appendix A). However, while strains belonging to the four phyla were isolated in both media, in M600, isolates belonged to 21 different genera, while in MA, isolates belonged to 18 genera (Appendix A). Moreover, M600 isolates showed higher diversity at the phylum level: *Actinomycetota* (10 in M600, 5 in MA) and *Bacteroidota* (three in M600, two in MA). On the other hand, MA isolates showed greater diversity in the phyla *Bacillota* (one in M600, three in MA) and *Pseudomonadota* (seven in M600, eight in MA) (Appendix A). Furthermore, while genera such as *Streptomyces* were equally isolated in both media, the isolates of *Corynebacterium*, *Dietzia*, *Kocuria*, *Microbacterium*, *Micromonospora*, *Nocardia*, *Aquimarina*, *Limimaricola*, *Marinobacter* and *Phaeobacter* were only retrieved in medium M600, evidencing the greater diversity obtained in this medium (Appendix A). Yet, genera such as *Nocardiopsis* appeared to prefer MA (23 isolates compared to the 6 in M600).

The genetic differentiation of closely related strains was assessed using BOX-PCR and ERIC-PCR and a 20% difference threshold for separation between genotypes [35] was applied. The results obtained show a clear grouping of strains into the different species of bacteria (Appendix A). Examples of bacterial strains with similar genotypes include *Arthrobacter gandavensis* PMIC_1E12B, PMIC_1F10B.1, PMIC_1F10C.1 and PMIC_2F9 (Appendix A) and strains with different genotypes such as *Streptomyces ardesiacus* PMIC_2D8A, PMIC_2D8B, PMIC_1C8A, PMIC_2C8A and PMIC_2C8B (three identified genotypes) (Appendix A). While in some cases there was a perfect match between the BOX-PCR and the ERIC-PCR fingerprinting, that was not the case in others, such as the strains of *Nocardiopsis prasina* (BOX-PCR identified two genotypes while ERIC-PCR identified three different genotypes) (Appendix A), *Nocardiopsis alba* (seven genotypes in the BOX-PCR and two in the ERIC-PCR) (Appendix A), *Streptomyces hydrogenans* (BOX-PCR did not identify any different genotypes, while three were identified in ERIC-PCR) (Appendix A) and *Streptomyces xiamenensis* (four identified in BOX-PCR, and two in ERIC-PCR) (Appendix A).

### 3.2. Antimicrobial Screening

All the strains of *Actinomycetota* were analysed for the putative presence of genes of NRPS and PKS-I enzyme complexes. Out of the 96 strains screened, a total of 53 strains with either one or both NRPS and PKS-I genes were detected (Table 1 and Figure 3). However, only three were positive for both genes (*S. ardesiacus* strain PMIC_2C8B, *R. erythropolis* strain PMIC_1E9B and *Rhodococcus coprophilus* strain PMIC_2E10) and only four presented the NRPS gene (Table 1 and Figure 3) (*Kocuria polaris* strain PMIC_1H7A, *Streptomyces albidoflavus* PMIC_1C12A, *S. xiamenensis* PMIC_2C2B and *Rhodococcus qingshengii* strain PMIC_2E9C). The PKS-I gene was the most widespread among the isolated strains, with 23 *Nocardiopsis*, 18 *Streptomyces*, 2 *Arthrobacter* and 1 *Nocardia*, *Plantibacter* and *Rhodococcus* strains, in a total of 46 strains (Table 1 and Figure 3). The 43 strains that did not amplify any of the genes (Table 1 and Figure 3) belonged to the genera *Streptomyces* (27), *Nocardiopsis* (6), *Microbacterium* (3, including the possible novel taxon, PMIC_1C1B), *Arthrobacter* (2), *Dietzia* (2), *Corynebacterium* (1), *Micromonospora* (1) and *Rhodococcus* (1).

As it has been shown that oligotrophy induces greater bioactivity in *Actinomycetota* [44], antimicrobial screening with the strains that putatively amplified at least one PKS-I or one NRPS gene was conducted with the extracts of these strains in medium M607. This medium is comparatively more oligotrophic than the isolation media MA or M600. As the bioactive compounds are products of secondary metabolism, the strains were allowed to grow for 15 days. This guarantees that all attained the stationary growth phase.

Of the 53 tested strains, only 12 strains were bioactive, with values above 50% growth inhibition (Table 1). Ten strains showed activity only against *E. coli* ATCC 25922, and one was bioactive only against *S. aureus* ATCC 29,213 (Table 1). *Streptomyces flavoviridis* strain PMIC_1A8B was bioactive against both targets, and all the bioactive strains putatively possessed genes associated with the PKS-I cluster (Table 1).

The one strain many compounds (OSMAC) [45] approach was performed with the 12 bioactive strains in media of varying levels of oligotrophy, 1:10 M607, M607, M600, MA and CGY (Table 2). Regarding *E. coli* ATCC 25922, some extracts did not show bioactivity, namely *A. gandavensis* PMIC_1E12B, *N. alba* PMIC_1A11B.2 and PMIC_1F6A.3, *R. coprophilus* PMIC_1E10C and *S. albidoflavus* PMIC_2C12. For strains *N. alba* PMIC_2A11B.1, *S. flavoviridis* PMIC_1A8B, *S. griseoflavus* PMIC_1D9B and *S. hydrogenans* PMIC_1I1A, a decrease in the degree of bioactivity compared to the first screening was observed, but the growth inhibition values were near the bioactivity threshold. Increased growth inhibition was verified for *N. nova* PMIC_1A10B, especially in the extracts from media 1:10 M607 (100%) and CGY (87%), and *S. setonii* in extracts from media 1:10 M607 (93.1%) and M607 (70.4%). *N. alba* PMIC_2H2C.2, which in the first screening showed growth inhibition values below the threshold (34.4%), showed an increase in bioactivity inhibition, especially in medium 1:10 M607 (91.7% inhibition). In relation to *S. aureus* ATCC 29213, 1:10 M607, M607 and M600 extracts from *S. flavoviridis* PMIC_1A8B showed very high inhibition values (100.0%, 77.2%, and 100.0%, respectively) (Table 2). Furthermore, *S. griseoflavus* PMIC_1D9B only became bioactive in CGY medium (91.1% inhibition), *S. hydrogenans* PMIC_1I1A became bioactive in 1:10 M607 medium (73.2% inhibition), and *N. alba* PMIC_1F6A.3 lost bioactivity.

All of the bioactive extracts were dereplicated (Appendix A), and the results from the initial and the OSMAC screenings are presented in Table 3. Extract dereplication is a critical step in natural product discovery, as it putatively detects and identifies known molecules in this early stage of the screening process. Certain compounds were often putatively found in the dereplications. This was the case of the diketopiperazines cyclo(L-Leu-L-Pro) (almost always present) and cyclo(Pro-Tyr), and the derivative of tyramine, N-acetyltyramine, with anti-*Vibrio anguillarum* [46], antifungal [47] and antimalaria activity [48], respectively. Germicidin A, a pyranone autoregulator of sporulation [49], was putatively present in the extracts from *N. alba* (PMIC_1A11B.2 and PMIC_2A11B.1). The antibiotic X-14952B, a 20-membered macrolide lactone with broad-spectrum antibacterial properties [50,51], was found in the extracts of *N. alba* PMIC_2H2C.2 and PMIC_2A11B.1. Also present in the extract of *N. alba* PMIC_1F6A.3 was cyclo(Tyr-Leu), a diketopiperazine that exhibits antibacterial [52] and antifungal [53] activities. Germicidin G and surugamide A were putatively present in *S. albidoflavus* PMIC_2C12 and *S. hydrogenans* PMIC_ I1A. Just like germicidin A, germicidin G also regulates spore germination [54,55], and surugamide A is a cyclic octapeptide with anticancer properties [56]. Ansalactam A, putatively present in *S. albidoflavus* PMIC_2C12 and *S. griseoflavus* PMIC_1D9B extracts, is an ansa macrolide with possible antibacterial activity [57] that was first isolated from a marine-sediment-derived bacterium of the genus *Streptomyces* [58]. 3-acetylamino-N-2-thienylpropanamide, with reported cytotoxic activity [59], was consistently found in *S. griseoflavus* PMIC_1D9B extracts. The antibiotic MKN-003B was present in all of the extracts from *S. hydrogenans* PMIC_1I1A. This strain also putatively produces surugamide E, blastomycin, antimycin A13 and antimycin A11. MKN-003B is a lactone with reported antifungal activity [60], although Lacret et al. [61] demonstrated no antifungal activities against *Candida albicans* MY1055 and *Aspergillus fumigatus* ATCC46645 and no antibacterial activity against methicillin-resistant *S. aureus* MB5393 and *E. coli* MB2884. Surugamide E, like surugamide A, is a cyclic octapeptide with cytotoxic activity, [56] and blastomycin is a polyene with fungicide properties [62]. Antimycins A11 and A13 are macrodiolides with potent and mild antifungal activity, respectively [63]. The extracts of *S. setonii* strain PMIC_1F12B showed the presence of corynecin I and chloramphenicol. Chloramphenicol is a dichloro-substituted acetamide with broad-spectrum antibacterial activity [64]. Corynecin I is a chloramphenicol-like acyl-nitrophenylpropylamine that also has broad-spectrum antibacterial activity [65,66].

In the various dereplications of the bioactive extracts, several non-identified molecules were found, for which the chemical formulae are provided in Table 3. Some formulae did not match any known compound in the DNP, namely C_9_H_10_N_2_O, C_12_H_25_NO_3_, C_14_H_29_NO_3_, C_16_H_24_O_3_, C_20_H_31_NO_4_S, C_22_H_44_O_12_, C_23_H_13_ClO_4_S_2_, C_24_H_48_O_13_, C_27_H_53_N_5_O_10_ and C_27_H_54_N_10_O_10_. Others matched more than one compound, although they might also constitute new natural products. This is the case of the following formulae: (1) C_20_H_13_N_3_O_6_ was found to match two compounds, the antibiotics A 33,853 and U 60394, both active against Gram-positive bacteria. (2) C_13_H_22_O_3_ matched with 10 coincidences in the DNP. Of these, 5-(6-hydroxy-7-methyloctyl)-2(5h)-furanone, trihomononactic acid lactone, 7-ethyl-10-hydroxy-7-undecene-3,6-dione, 7-ethyl-9-hydroxy-7-undecene-3,6-dione and 7-ethyl-4-hydroxy-7-undecene-3,6-dione showed no known antimicrobial activity. Acaterin, 3-(1-hydroxyoctyl)-5-methyl-2(5H)-furanone is an inhibitor of acyl-cholesterol acyltransferase [67]. MKN 003C, a butenolide containing a branched side chain very similar to MKN-003B, showed antifouling activity [60]. 5-(6-hydroxy-6-methyloctyl)-2(5H)-furanone showed anti-adenoviral activity, while 5-(7-hydroxy-6-methyloctyl)-2(5H)-furanone and 5-(6-methyloctyl)-2(5H)-furanone, (5S,6′S)-form, 7′S-hydroxy, showed cytotoxic activity [68]. (3) C_15_H_24_O_3_ matched with nine molecules in the DNP (Table 3). 10,15-dihydroxy-4-cadinen-3-one is a sesquiterpene [69], nocapyrone R is an α-pyrone [70], nocardiopyrone A is also an α-pyrone [71], presulficidin C is a triketide pyrone [72], and streptoone C is a ketonic acid [73], all with no known relevant biological importance. Kandenol A is a sesquiterpene with weak antimicrobial activities against *Bacillus subtilis* and *Mycobacterium vaccae* [74], nocapyrone H is an α-pyrone with anti-inflammatory effects [75], marinactinone A is a γ-pyrone with cytotoxic activity [76] and photopyrone A is also an α-pyrone that plays a role in quorum-sensing signalling for *Photorhabdus luminescens* [77]. (4) C_15_H_24_O_4_ matched with 10 different molecules in the DNP. Of these, 1,3,5-bisabolatriene-1,7,11,15-tetrol and 2,6-farnesadiene-1,12-dioic acid, presulficidin D [72], and nocapyrones A, I, J, N and O [78,79] have no known biological importance. Kandenol B and D are sesquiterpenes with weak antimicrobial activities against *B. subtilis* and *M*. *vaccae* [74].

## 4. Discussion

The beach sand microbiota (micropsammon, the microorganisms inhabiting supratidal and intertidal sand [80]) has been poorly studied [81]. This environment seems to be dominated by *Pseudomonadota* and *Bacteroidota* [82,83,84], as also shown in a study of coastal sediments in three sites in Quanzhou Bay, Fujian Province, China, where Huang et al. [85] isolated 1036 bacterial strains, most of which were *Pseudomonadota* (75.77%), and only 13.32% were *Actinomycetota*. However, in our study with the modified iChip methodology and without any especial selective treatment, a much higher number of *Actinomycetota* (60.8%) compared to the lower percentage of *Pseudomonadota* (20.9%) and Bacteroidota (4.4%) was obtained. This methodology also allowed the obtainment of a great diversity at genus level (29 different genera). About the same number of isolates were obtained in the two media used (Figure 3 and Appendix A), and a higher diversity at genus level was obtained in M600 compared with medium MA. These results may indicate that the nutritional formulation of M600 might be more adequate to isolate the diverse microbiological community present in marine sediments. In fact, the composition of M600 includes Hutner’s basal salts [86] and a vitamin solution [87] as supplements that are not available in MA medium. Vitamins are essential for many microorganisms that are unable to synthetize them. Furthermore, some micronutrients, such as Fe^2+^, which can be used as alternative energy sources [88], limit the growth of microorganisms if they are deficient in the culture medium.

With the iChip in the present study, and considering only the *Actinomycetota*, 50% belong to the genus *Streptomyces* and 30.2% to *Nocardiopsis*. The iChip approach allowed for the isolation of a great quantity of *Streptomyces* and *Nocardiopsis*, while only one *Micromonospora*-related isolate and no *Salinispora* strains were obtained. Other studies were also focused exclusively on the isolation of *Actinomycetota*. Ribeiro et al. [89], whose aim was to specifically obtain sporogenous *Actinomycetota* from marine sediments in Cepães beach, Esposende (north of Memoria beach), used three different media, two of which were selective for *Actinomycetota*. With these media, out of a total of 52 actinobacterial isolates, 67% belonged to the genus *Micromonospora*, 17% to *Streptomyces* and 8% to *Arthrobacter.* In a study of the coast of Diu Island, Gujarat, India, inter-tidal sediments were dried in a laminar flow hood for 16 h and isolation media were supplemented with cycloheximide and nalidixic acid. In the study, the authors isolated 148 *Actinomycetota* strains, of which *Streptomyces* spp. accounted for 61.5%, *Micromonospora* spp. 15.4%, *Nocardiopsis* spp. 10.2%, *Saccharomonospora* spp. 5.1%, *Actinomadura* spp. 2.6%, *Glycomyces* spp. 2.6% and *Nocardia* spp. 2.6% [90]. In another isolation experiment focused also only on *Actinomycetota*, Prieto-Davo et al. [91] isolated a total of 400 actinobacterial strains from near-shore and off-shore sediments in the Madeira archipelago, the majority belonging to *Streptomyces* (44.5%), *Micromonospora* (22.5%), *Salinispora* (8.5%) and *Nocardiopsis* (7%). *Salinispora* spp. are believed to inhabit only tropical and subtropical regions (to which the Madeira archipelago belongs) of the oceans, where thousands of strains have been isolated [92,93,94], while Memoria beach is in a temperate zone and bathed by cold Atlantic waters. Thus, this may be the reason for the absence of isolates from this genus in the present study. Unlike any of the previous isolation attempts, no particular pre-treatments (use of heat, dryness of sediments or antibiotics) were necessary to isolate *Actinomycetota* in the iChip isolation attempt. Additionally, the agar medium plates did not require any treatment with antibiotics to minimize the development of fast-growing bacteria, as the suspension placed in the wells usually contained only one or two organisms (Appendix A). This technique greatly reduced and simplified the isolation process and allowed the obtainment of a high diversity of bacterial genera and species.

The detection of the presence of secondary-metabolite-associated genes, such as PKS-I or NRPS genes, is an important first step to assess the biotechnological potential of the strains under study [6,41]. This molecular analysis showed that about 55% of the tested strains putatively possess one or both genes and allowed our antimicrobial screening to focus on these strains. In the comparatively more oligotrophic M607 medium, 12 strains (five *Streptomyces*, four *Nocardiopsis*, one *Arthrobacter*, one *Nocardia* and one *Rhodococcus*) were bioactive.

The literature reveals how relevant these genera are in the discovery of unique and biotechnologically useful organic molecules [6]. Subsequently, the OSMAC screening shows that some extracts no longer showed bioactivity. While the loss of bioactivity is difficult to justify, a factor that may be determinant is the composition of the natural seawater used in media formulations, the composition of which varies over time. Moreover, peptone and yeast extracts may also play a role, as the exact chemical compositions of peptone and yeast extracts are not fully known and may have specific signalling molecules interfering with bioactive compound production [95,96].

Data on the bioactivity of strains from the genus *Arthrobacter* are still limited. While these strains are present and abundant in many environments, soils and sediments appear to be their preferred habitat. The genus is an industrially relevant bioactive group and a source of glutamic acid, α-ketoglutaric acid and riboflavin [97]. Furthermore, Rojas et al. [40] showed that two strains of *Arthrobacter agilis* isolated from biological mats from the Antarctic region produced bioactive cyclic thiazolyl peptides that proved to be effective against Gram-positive bacteria, in particular, *S. aureus* ATCC 29213. Our results show that *Arthrobacter gandavensis* PMIC_1E12B produced an extract with bioactive properties against *E. coli* ATCC 25922, which could be due to the presence of two diketopiperazines (Table 2), but not against *S*. *aureus* ATCC 29213.

The bacteria from the genus *Nocardia* are a well-known source of bioactive natural products, and in the review by Dhakal et al. [98], 47 major bioactive molecules isolated from *Nocardia* spp. were described, showing a vast chemical diversity from which 26 molecules showed antimicrobial activity. Curiously, regarding antibacterial or other antimicrobial activities, none appeared to be associated with strains affiliated with *N. nova*, a genus that was bioactive in our study. The anti-*E*. *coli* ATCC 25,922 bioactivity observed by *N*. *nova* PMIC_1A10B may be due to the putative presence of cyclo(L-Leu-L-Pro), cyclo(Pro-Tyr) or six unidentified molecules. The results herein show that this strain putatively possesses a great capacity for producing molecules with antimicrobial properties and may point to the presence of novel natural products.

Regarding the genus *Nocardiopsis*, reports on the production of natural products with antimicrobial properties are abundant [99,100]. *Nocardiopsis*-affiliated strains have been shown to produce various compounds, such as the phenazines 1,6-dihydroxyphenazine and 1,6-dihydroxyphenazine 5,10-dioxide, which were isolated from the marine strain OPC-15, affiliated with *Nocardiopsis dassonvillei* [101]. These phenazines showed antimicrobial activity against *Proteus mirabilis* and *B. subtilis*. From a marine-derived *N. alba*, the thiopeptide TP-1161 was isolated [102]. TP-1161 displayed broad-spectrum antibacterial activity against *Staphylococcus haemolyticus*, *Staphylococcus epidermidis*, *S. aureus, Streptococcus pneumoniae*, *Enterococcus faecalis* and *Enterococcus faecium*. The production by *N. alba* of several diketopiperazines that possess antibacterial properties has been reported [103,104]. For example, albonoursin and methoxylated albonoursin, isolated from *N. alba* strain ATCC BAA-2165, have been shown to inhibit the growth of the bee pathogen *Paenibacillus larvae* [105]. Moreover, a novel derivative of 3-acetyl-dimethyl sterol, which was isolated from a marine actinomycetes *N. alba* MCCB 110, showed promising antibacterial activity against the shrimp aquaculture pathogen *Vibrio harveyi* [106]. In fact, the dereplication of the extracts of the four strains of *N. alba* of the present study consistently showed the presence of diketopiperazines (Table 3). Furthermore, the presence of N-acetyltyramine and the antibiotic X-14952B may explain the results obtained for extracts from the CGY and M600 media. Curiously, 1:10 M607 extract showed only the presence of cyclo(L-Leu-L-Pro), yet it proved to be the most bioactive. Several non-identified molecules, such as C_15_H_24_O_4_, C_16_H_24_O_3_ and C_27_H_52_N_10_O_10_, may also explain the bioactivity due to *N. alba* strains. The results presented here show the capability of the genus *Nocardiopsis* to produce relevant bioactive compounds, pointing to the presence of novel natural products.

Although strains PMIC_1A11B.2, PMIC_2A11B.1, PMIC_1F6A.3 and PMIC_2H2C.2 are all 100% phylogenetically affiliated with *N. alba* DSM 43377, ERIC-PCR and BOX-PCR fingerprinting showed that they belong to two different genotypes. With the ERIC-PCR, PMIC_1A11B.2, PMIC_2A11B.1 and PMIC_1F6A.3 were grouped in one genotype and PMIC_2H2C.2 in another, while in the BOX-PCR, PMIC_1A11B.2 and PMIC_2A11B.1 were grouped in one genotype and PMIC_1F6A.3 and PMIC_2H2C.2 in a second. The bioactive profile of the first extract also showed differences regarding the strains, with strains PMIC_1A11B.2, PMIC_2A11B.1 and PMIC_1F6A.3 showing high inhibition of *E. coli* ATCC 25922, and strain PMIC_2H2C.2 showing mild anti-*S. aureus* ATCC 29,213 activity, which, interestingly, coincides with the ERIC-PCR fingerprinting cluster results.

Data regarding the genus *Rhodococcus* show its biotechnological potential. *Rhodococcus*-affiliated strains have been shown to produce lariatins, cyclic peptides with selected bioactivity against *Mycobacterium tuberculosis* [107], and rhodostreptomycins, aminoglycosides with bioactivity against *E*. *coli*, *S*. *aureus* and *B*. *subtillis* [108]. However, no data are available on the bioactive potential of *Rhodococcus coprophilus*. The strain tested here, *R*. *coprophilus* PMIC_1E10C, showed very strong anti-*E. coli* activity (Table 2), which may be due to the putative presence of the bioactive diketopiperazine cyclo(l-leu-l-pro) and an unidentifiable molecule with the formula C_14_H_29_NO_3_. Further testing with this strain may prove useful in determining additional bioactivities and maybe even novel natural products, as there is a clear gap in the knowledge within this genus, which has prime biotechnological potential.

The genus *Streptomyces* is the gold standard of natural products and drug discovery research, with more than 6000 different compounds being described with an origin in this genus [109]. Moreover, strains closely associated with the ones obtained with the iChip approach have already been shown to be reservoirs of biotechnologically interesting molecules. For example, *S. albidoflavus* strain I07A-01824 was shown to produce Antimycin A_18_, an effective antifungal agent [110]. For *S. flavoviridis* ATCC 21892, abundant data show its ability to produce natural products with antimicrobial properties [111]. An example is zorbamycin, a glycopeptide antibiotic first reported in 1971 that can rapidly induce the degradation of deoxyribonucleic and ribonucleic acids in *B. subtilis* and *E. coli* [112]. *S*. *griseoflavus* is also a well-known producer of natural products, such as bicozamycin, an antibacterial agent that inhibits the Rho protein of *E. coli* [113], and colabomycins A-C, which have anti-Gram-positive bacteria and cytotoxic activities [114]. *S. hydrogenans* strain KMFA-1 was shown to produce extracts with antifungal properties [115]. Moreover, Lewis et al. [116] recently showed that strains closely related to *S. hydrogenans* and *S. albidoflavus* produce the polyene antifungal candicidin. *S. setonii* strains also have antifungal properties, due to the compound FR109615, an aminocyclopentane isolated from *S. setonii* strain no. 7562 [117,118].

The bioactivity found in the *Streptomyces* strains detected in this work may be justified by the action of molecules such as diketopiperazines, N-acetyltyramine, antibiotic MKN-003B, germicidin G, surugamide A and E, blastmycin, antimycins A11 and A13, ansalactam A, corynecin I and chloramphenicol, as well as non-identified molecules such as C_13_H_22_O_3_, C_20_H_13_N_3_O_6_, C_22_H_44_O_12_, C_27_H_53_N_5_O_10_ and C_23_H_13_ClO_4_S_2_. This last molecule seems to be of special interest, as the presence of a halogen (chlorine) and two sulphur atoms may indicate potential bioactivity [119,120]. Overall, these results suggest the relevance of the genus *Streptomyces* for the discovery of natural products and drug discovery research, as the extracts of the strains studied here showed antibacterial activity and putatively contain bioactive compounds and possible novel natural products.

## 5. Conclusions

The in situ conditions simulated by the use of an iChip-inspired methodology allowed the recovery of a large number of diverse bacteria from marine sediments, which is demonstrated by the high number of overall domesticated species and the number of genotypes that were identified using PCR fingerprinting techniques. Moreover, a putative novel taxon was isolated (98.48% 16S rRNA gene similarity to the closest known strain). Regarding the bioactive potential of the isolated actinobacterial strains, a high number possessed genes associated with bioactive potential, and 12 strains had at least one inhibitory effect on *E. coli* ATCC 25,922 and/or *S. aureus* ATCC 29213. The dereplication of their extracts showed the putative presence of several known bioactive molecules, and of some with unknown identity that might constitute new natural antibiotics. In conclusion, the highly diverse *Actinomycetota* isolated in this study have great biosynthetic potential and, thus, may prove to be useful biotechnological tools.

## Figures and Tables

**Figure 1 microorganisms-10-01471-f001:**
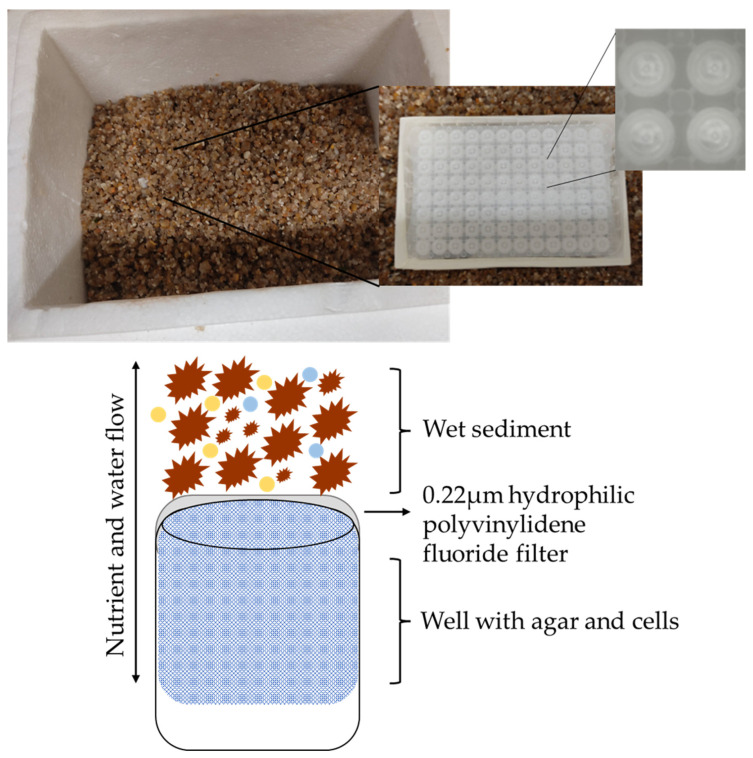
Scheme of the iChip methodology using the MultiScreen^®^ 96-Well Filtration Plate (Nunc, Roskilde, Denmark) in a sediment box to simulate the natural environment.

**Figure 2 microorganisms-10-01471-f002:**
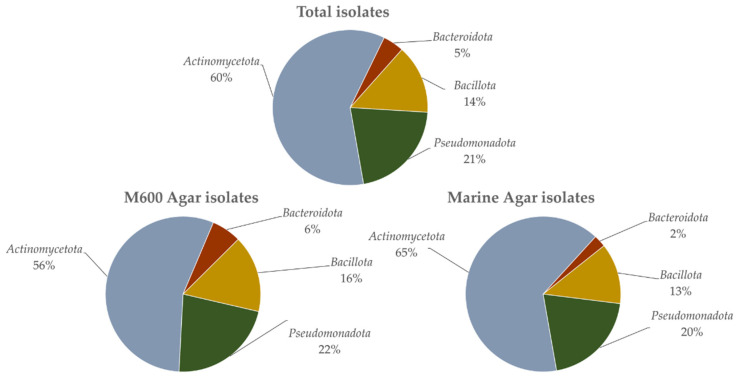
Distribution of isolated strains by phylum in total numbers and by isolation medium (M600 and MA).

**Figure 3 microorganisms-10-01471-f003:**
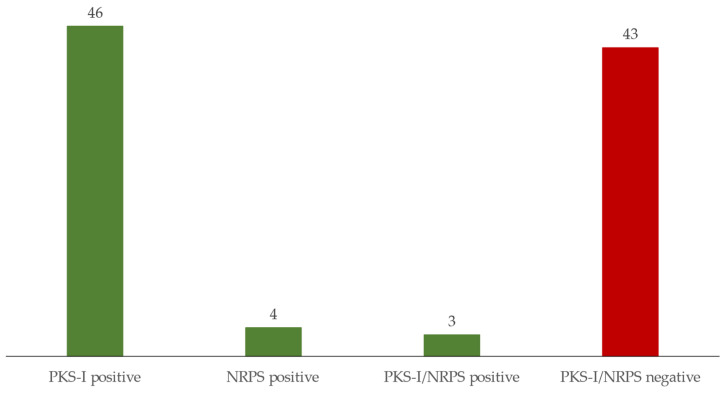
Number of isolated *Actinomycetota* strains putatively presenting PKS-I and NRPS genes.

**Table 1 microorganisms-10-01471-t001:** Presence of secondary metabolism genes in Actinobacterial strains and mean inhibition of the extracts produced from the first extract.

Strain ID	Affiliation		NRPS	PKS-I	% Inhibition	
	%	Taxonomic Unit			*E*. *coli*	*S*. *aureus*
PMIC_1E12B	100.00	*Arthrobacter gandavensis* R 5812	N.A	A	64.8	39.6
PMIC_1F10C.1	100.00	*Arthrobacter gandavensis* R 5812	N.A	A	1.7	2.9
PMIC_1H7A	100.00	*Kocuria polaris* CMS 76or	A	N.A	1.9	−16.3
PMIC_1A10B	99.82	*Nocardia nova* NBRC 15556	N.A	A	50.6	−12.7
PMIC_1A11B.2	100.00	*Nocardiopsis alba* DSM 43377	N.A	A	76.3	17.6
PMIC_1F6A.3	99.76	*Nocardiopsis alba* DSM 43377	N.A	A	79.2	22.4
PMIC_2A11A.1	99.92	*Nocardiopsis alba* DSM 43377	N.A	A	39.6	19.2
PMIC_2A11B.1	100.00	*Nocardiopsis alba* DSM 43377	N.A	A	77.7	−8.3
PMIC_2A11B.2	100.00	*Nocardiopsis alba* DSM 43377	N.A	A	12.9	4.2
PMIC_2A11B.3	100.00	*Nocardiopsis alba* DSM 43377	N.A	A	37.4	22.6
PMIC_2F6A	99.76	*Nocardiopsis alba* DSM 43377	N.A	A	45.2	30.8
PMIC_2F6B	99.76	*Nocardiopsis alba* DSM 43377	N.A	A	15.7	1.0
PMIC_2F6C	99.77	*Nocardiopsis alba* DSM 43377	N.A	A	19.4	18.6
PMIC_2H2A	100.00	*Nocardiopsis alba* DSM 43377	N.A	A	26.7	26.8
PMIC_2H2C.1	100.00	*Nocardiopsis alba* DSM 43377	N.A	A	38.7	48.5
PMIC_2H2C.2	100.00	*Nocardiopsis alba* DSM 43377	N.A	A	33.4	52.0
PMIC_2B1A	99.85	*Nocardiopsis prasina* DSM 43845	N.A	A	14.1	3.4
PMIC_2B1C	99.82	*Nocardiopsis prasina* DSM 43845	N.A	A	12.8	17.0
PMIC_2B1D	99.85	*Nocardiopsis prasina* DSM 43845	N.A	A	38.4	49.4
PMIC_2D10A	99.84	*Nocardiopsis prasina* DSM 43845	N.A	A	20.0	14.9
PMIC_2D10B.1	99.75	*Nocardiopsis prasina* DSM 43845	N.A	A	35.6	22.2
PMIC_2D10B.2	99.84	*Nocardiopsis prasina* DSM 43845	N.A	A	19.8	7.5
PMIC_2D10C	99.84	*Nocardiopsis prasina* DSM 43845	N.A	A	44.1	36.1
PMIC_2F12A	99.82	*Nocardiopsis prasina DSM 43845*	N.A	A	−5.4	17.3
PMIC_2C3B.2	99.11	*Nocardiopsis umidischolae* 66/93	N.A	A	41.4	7.9
PMIC_2C3B.3	99.19	*Nocardiopsis umidischolae* 66/93	N.A	A	39.1	23.1
PMIC_2C3B.4	99.20	*Nocardiopsis umidischolae* 66/93	N.A	A	16.9	13.5
PMIC_2H6	99.81	*Plantibacter flavus* VKM Ac-2504	N.A	A	−3.4	−9.8
PMIC_1E10C	99.69	*Rhodococcus coprophilus* NBRC 100603	N.A	A	86.7	16.6
PMIC_2E10	99.69	*Rhodococcus coprophilus* NBRC 100603	A	A	24	−9.5
PMIC_1E9B	100.00	*Rhodococcus erythropolis* NBRC 15567	A	A	8.9	−0.9
PMIC_2E9C	99.76	*Rhodococcus qingshengii* JCM 15477	A	N.A	3.2	−11.5
PMIC_1C12A	99.67	*Streptomyces albidoflavus* DSM 40455	A	N.A	1.9	−18.2
PMIC_2C12	99.69	*Streptomyces albidoflavus* DSM 40455	N.A	A	64.8	13.1
PMIC_1A8C	99.66	*Streptomyces albogriseolus* NRRL B-1305	N.A	A	2.0	39.9
PMIC_2G8C	99.68	*Streptomyces ambofaciens* ATCC 23877	N.A	A	30.9	12.0
PMIC_1C8A	99.69	*Streptomyces ardesiacus* NRRL B-1773	N.A	A	34.0	12.3
PMIC_2C8A	99.69	*Streptomyces ardesiacus* NRRL B-1773	N.A	A	33.1	31.9
PMIC_2C8B	99.70	*Streptomyces ardesiacus* NRRL B-1773	A	A	10.7	−2.0
PMIC_2D8A	100.00	*Streptomyces ardesiacus* NRRL B-1773	N.A	A	20.8	10.4
PMIC_2D8B	100.00	*Streptomyces ardesiacus* NRRL B-1773	N.A	A	27.6	28.4
PMIC_1A8B	99.43	*Streptomyces flavoviridis* NBRC 12772	N.A	A	75.9	100
PMIC_1D9A	99.80	*Streptomyces griseoflavus* LMG 19344	N.A	A	30.1	4.8
PMIC_1D9B	99.75	*Streptomyces griseoflavus* LMG 19344	N.A	A	64.6	9.8
PMIC_1I1A	100.00	*Streptomyces hydrogenans* NBRC 13475	N.A	A	80.2	11.9
PMIC_1I1B	100.00	*Streptomyces hydrogenans* NBRC 13475	N.A	A	30.1	26.6
PMIC_2C8C	99.57	*Streptomyces hydrogenans* NBRC 13475	N.A	A	36.5	15.6
PMIC_2D11A.2	99.69	*Streptomyces hydrogenans* NBRC 13475	N.A	A	46.1	31.7
PMIC_2D11C	99.70	*Streptomyces hydrogenans* NBRC 13475	N.A	A	21.4	26.1
PMIC_1F12B	100.00	*Streptomyces setonii* NRRL ISP-5322	N.A	A	62.1	21.6
PMIC_2F12B	100.00	*Streptomyces setonii* NRRL ISP-5322	N.A	A	28.5	40.8
PMIC_1B3A.1	99.52	*Streptomyces xiamenensis* MCCC 1A01550	N.A	A	29.4	15.5
PMIC_2C2B	99.81	*Streptomyces xiamenensis* MCCC 1A01550	A	N.A	35.8	18.8
Culture medium extract					25.8	9.9

A = Amplified; N.A = Not Amplified.

**Table 2 microorganisms-10-01471-t002:** Bioactive actinomycetotal strains’ average inhibition of the extracts produced from all fermentations.

Strain ID	Affiliation		Inhibition (%)						Inhibition (%)					
			*E. coli*						*S. aureus*					
	%	Taxonomic Unit	M607	OSMAC 1:10 M607	OSMAC M607	OSMAC M600	OSMAC MA	OSMAC CGY	M607	OSMAC 1:10 M607	OSMAC M607	OSMAC M600	OSMAC MA	OSMAC CGY
PMIC_1E12B	100%	*Arthrobacter gandavensis* R 5812	64.8	24.6	17.4	16.3	29.8	29.9	39.6	−6.5	−9.8	3.7	2.8	−10.4
PMIC_1A10B	99.82%	*Nocardia nova* NBRC 15556	50.6	100.0	62.0	79.9	54.3	87.0	−12.7	28.8	27.0	22.6	−25.4	6.2
PMIC_1A11B.2	100%	*Nocardiopsis alba* DSM 43377	76.3	23.6	17.4	16.3	29.8	29.9	17.6	−10.1	−9.8	3.7	2.8	−10.4
PMIC_2A11B.1	100%	*Nocardiopsis alba* DSM 43377	77.7	40.9	43.1	45.7	41.7	45.0	−8.3	19.2	29.0	19.0	16.7	36.5
PMIC_1F6A.3	99.76%	*Nocardiopsis alba* DSM 43377	79.2	14.7	21.4	15.4	26.1	15.9	22.4	−12.2	−13.5	−16.8	9.5	−3.2
PMIC_2H2C.2	100%	*Nocardiopsis alba* DSM 43377	33.4	91.7	42.9	54.7	44.1	68.1	52.0	26.5	30.9	40.3	31.7	41.1
PMIC_1E10C	99.69%	*Rhodococcus coprophilus* NBRC 100603	86.7	31.2	44.9	48.0	42.7	33.1	16.6	23.9	24.8	7.6	19.3	21.0
PMIC_2C12	99.69%	*Streptomyces albidoflavus* DSM 40455	64.8	22.0	35.4	31.0	32.2	0.0	13.1	19.8	−14.6	−9.2	−3.7	−15.3
PMIC_1A8B	99.43%	*Streptomyces flavoviridis* NBRC 12772	75.9	39.1	41.8	40.5	42.4	46.7	100.0	100.0	77.2	100.0	37.0	40.3
PMIC_1D9B	99.75%	*Streptomyces griseoflavus* LMG 19344	64.6	57.2	54.3	54.2	43.2	98.3	9.8	27.8	34.3	27.3	5.0	91.1
PMIC_1I1A	100%	*Streptomyces hydrogenans* NBRC 13475	80.2	53.7	44.0	49.7	44.9	52.0	11.9	73.2	14.8	14.3	19.1	17.0
PMIC_1F12B	100%	*Streptomyces setonii* NRRL ISP-5322	62.1	93.1	70.4	52.9	48.7	42.1	21.6	31.5	33.5	21.9	23.3	−15.0
Culture medium extract			25.8	9.0	11.7	15.4	9.3	−0.2	9.9	14.7	12.0	10.2	5.5	5.9

**Table 3 microorganisms-10-01471-t003:** Putatively identified bioactive compounds and non-identified molecules present in the extracts.

Strain ID	Taxonomic Unit	Medium	Putatively Detected Bioactive Molecules	Non-Identified Molecules
PMIC_1E12B	*Arthrobacter gandavensis*	M607	cyclo(Pro-Tyr), cyclo(L-Leu-L-Pro)	−
		OSMAC 1:10 M607	−	−
		OSMAC M607	−	−
		OSMAC M600	−	−
		OSMAC MA	−	−
		OSMAC CGY	−	−
PMIC_1A10B	*Nocardia nova*	M607	N-acetyltyramine, cyclo(L-Leu-L-Pro)	−
		OSMAC 1:10 M607	N-acetyltyramine, cyclo(L-Leu-L-Pro)	C_9_H_10_N_2_O; C_15_H_24_O_3_
		OSMAC M607	cyclo(Pro-Tyr), cyclo(L-Leu-L-Pro)	C_27_H_54_N_10_O_10_
		OSMAC M600	cyclo(pro-tyr), N-acetyltyramine, cyclo(L-Leu-L-Pro)	C_22_H_44_O_12_
		OSMAC MA	cyclo(pro-tyr), N-acetyltyramine, cyclo(L-Leu-L-Pro)	−
		OSMAC CGY	N-acetyltyramine, cyclo(L-Leu-L-Pro)	C_22_H_44_O_12_; C_24_H_48_O_13_
PMIC_1A11B.2	*Nocardiopsis alba*	M607	cyclo(pro-tyr), N-acetyltyramine, cyclo(L-Leu-L-Pro), germicidin A	C_12_H_25_NO_3_, C_14_H_29_NO_3_
		OSMAC 1:10 M607	−	−
		OSMAC M607	−	−
		OSMAC M600	−	−
		OSMAC MA	−	−
		OSMAC CGY	−	−
PMIC_2A11B.1	*Nocardiopsis alba*	M607	cyclo(Pro-Trp), N-acetyltyramine, cyclo(L-Leu-L-Pro), germicidin A	−
		OSMAC 1:10 M607	−	−
		OSMAC M607	−	−
		OSMAC M600	N-acetyltyramine, cyclo(L-Leu-L-Pro), cyclo(Pro-Trp), germicidin A	−
		OSMAC MA	−	−
		OSMAC CGY	N-acetyltyramine, cyclo(L-Leu-L-Pro), germicidin A, antibiotic X-14952B	−
PMIC_1F6A.3	*Nocardiopsis alba*	M607	cyclo(Pro-Tyr), cyclo(Tyr-Leu), cyclo(L-Leu-L-Pro)	C_20_H_31_NO_4_S
		OSMAC 1:10 M607	−	−
		OSMAC M607	−	−
		OSMAC M600	−	−
		OSMAC MA	−	−
		OSMAC CGY	−	−
PMIC_2H2C.2	*Nocardiopsis alba*	M607	cyclo(L-Leu-L-Pro)	−
		OSMAC 1:10 M607	cyclo(L-Leu-L-Pro)	C_27_H_52_N_10_O_10_
		OSMAC M607	−	−
		OSMAC M600	N-acetyltyramine, cyclo(L-Leu-L-Pro), antibiotic X-14952B	C_15_H_24_O_4_
		OSMAC MA	−	−
		OSMAC CGY	cyclo(Pro-Tyr), N-acetyltyramine, cyclo(L-Leu-L-Pro), antibiotic X-14952B	C_15_H_24_O_4_, C_16_H_24_O_3_
PMIC_1E10C	*Rhodococcus coprophilus*	M607	cyclo(L-Leu-L-Pro)	C_14_H_29_NO_3_
		OSMAC 1:10 M607	−	−
		OSMAC M607	−	−
		OSMAC M600	−	−
		OSMAC MA	−	−
		OSMAC CGY	−	−
PMIC_2C12	*Streptomyces albidoflavus*	M607	cyclo(Pro-Tyr), N-acetyltyramine, cyclo(L-Leu-L-Pro), germicidin G, surugamide A, ansalactam A	−
		OSMAC 1:10 M607	−	−
		OSMAC M607	−	−
		OSMAC M600	−	−
		OSMAC MA	−	−
		OSMAC CGY	−	−
PMIC_1A8B	*Streptomyces flavoviridis*	M607	N-acetyltyramine, cyclo(L-Leu-L-Pro),	C_20_H_13_N_3_O_6_, C_23_H_13_ClO_4_S_2_
		OSMAC 1:10 M607	cyclo(L-Leu-L-Pro)	C_23_H_13_ClO_4_S_2_
		OSMAC M607	cyclo(pro-tyr), N-acetyltyramine, cyclo(L-Leu-L-Pro)	−
		OSMAC M600	N-acetyltyramine, cyclo(L-Leu-L-Pro)	C_20_H_13_N_3_O_6_, C_23_H_13_ClO_4_S_2_
		OSMAC MA	−	−
		OSMAC CGY	−	−
PMIC_1D9B	*Streptomyces griseoflavus*	M607	N-acetyltyramine, cyclo(L-Leu-L-Pro), ansalactam A	−
		OSMAC 1:10 M607	N-acetyltyramine, ansalactam A	−
		OSMAC M607	cyclo(Pro-Tyr), N-acetyltyramine, cyclo(L-Leu-L-Pro), 3-acetylamino-N-2-thienylpropanamide, ansalactam A	−
		OSMAC M600	cyclo(Pro-Tyr), N-acetyltyramine, cyclo(L-Leu-L-Pro), 3-acetylamino-N-2-thienylpropanamide, ansalactam A	−
		OSMAC MA	−	−
		OSMAC CGY	N-acetyltyramine, 3-acetylamino-N-2-thienylpropanamide, ansalactam A	−
PMIC_1I1A	*Streptomyces hydrogenans*	M607	cyclo(Pro-Tyr), N-acetyltyramine, cyclo(L-Leu-L-Pro), antibiotic MKN-003B, germicidin G, surugamide A	C_13_H_22_O_3_
		OSMAC 1:10 M607	cyclo(L-Leu-L-Pro), antibiotic MKN-003B, germicidin G, surugamide E, surugamide A	C_13_H_22_O_3_
		OSMAC M607	−	−
		OSMAC M600	−	−
		OSMAC MA	−	−
		OSMAC CGY	cyclo(Pro-Tyr), N-acetyltyramine, cyclo(L-Leu-L-Pro), antibiotic MKN-003B, germicidin G, surugamide E, surugamide A, blastmycin, antimycin A13, antimycin A11	C_13_H_22_O_3_
PMIC_1F12B	*Streptomyces setonii*	M607	N-acetyltyramine, corynecin I, chloramphenicol	−
		OSMAC 1:10 M607	N-acetyltyramine, cyclo(L-Leu-L-Pro)	C_27_H_53_N_5_O_10_
		OSMAC M607	−	−
		OSMAC M600	N-acetyltyramine, cyclo(L-Leu-L-Pro)	−
		OSMAC MA	−	−
		OSMAC CGY	N-acetyltyramine, cyclo(L-Leu-L-Pro)	C_22_H_44_O_12_

−: Not dereplicated.

## Data Availability

Not applicable.

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
