# Peer review of "iChip-Inspired Isolation, Bioactivities and Dereplication of Actinomycetota from Portuguese Beach Sediments"

_microorganisms, 2022, doi:10.3390/microorganisms10071471_

Round 1

Reviewer 1 Report

The manuscript provided an exploration of marine microorganisms (Actinomycetota. Marine sediments from Memoria beach (Portugal), were used for the isolation by the iChip. The isolates were identified, searched for the putative presence of secondary metabolism genes associated with polyketide synthase I (PKS-I) and non-ribosomal peptide synthetases (NRPS), screened for antimicrobial activity against Escherichia coli ATCC 25922 and Staphylococcus aureus ATCC 29213 then bioactive extracts were dereplicated by LC/HRMS. The obtained results indicated the use of the isolated bacteria as sources of new bioactive natural products.

The manuscript includes all the required parts.

To improve the quality of the document, I cite some recommendations:

- The proposed sections titles included in the Introduction part, could be avoided.

- The figures included in results part should be well presented (e.g. Table 3, Figure 2).

- Reference citation in the manuscript must be uniform and respecting journal guidelines (references included in discussion part).

- English revision is required for all the manuscript.

Author Response

Answers for Comments and Suggestions of Reviewer 1

Q. The proposed sections titles included in the Introduction part, could be avoided.

A. According to the journal guidelines, it is recommended to use sections titles but to please the reviewer those were removed.

Q. The figures included in results part should be well presented (e.g. Table 3, Figure 2).

A. To be easier to read the table horizontal lines were introduced in Table 3. Figure 2 was changed according to reviewer 3.

Q. Reference citation in the manuscript must be uniform and respecting journal guidelines (references included in discussion part).

A. Reference citations was revised.

Q. English revision is required for all the manuscript.

A. Minor English corrections were done

Reviewer 2 Report

Marine microbiota is showing great potential in developing novel bactericidal drugs, while the hardness of cultivation of marine bacteria limits its applications to develop marine microbial natural products. In this manuscript, the authors provide an iChip-based isolation method from sediments, which exhibits huge potential to mine new antibiotics and their producers. Below are my concerns about this manuscript:

Fig. 1

1. for the total192 isolates, how many isolates were from M600 or Marine agar, respectively?

2. How was the total phylum percentage calculated? Was a 16s rRNA high-throughput sequencing or just the average of M600 and marine agar bacteria?

Fig. 2

To avoid confusion, please keep the taxonomic names to be consistent among figures and texts. Such as Pseudomonadota was used in the text, but Proteobacteria was used in figures.

In line 243, Fig. 3 doesn’t show the data claimed in the text.

Lots of culture media were used in this manuscript, for the convenience of follow-up research from other labs, please provide the gradients of media, or catalogs if possible.

Please add a reference for OSMAC.

Though iChip is talented in primary enrichment of marine bacteria, the M600 or marine agar-based culture definitely would result in biased enrichment of specific types of bacteria, which may explain the massive difference in microbiota composition from different researches. Meta-genomic sequencing-based research after iChip enrichment may provide a more in-depth analysis of new bactericidal natural products and their producers.

Author Response

Answers for Comments and Suggestions of Reviewer 2

Q. for the total192 isolates, how many isolates were from M600 or Marine agar, respectively?

A. In total, we obtained 79 strains in M600 and in MA. In M600, 45 belonged to Actinomycetota, 17 to Pseudomonadota, 12 Bacillota and 4 Bacteroidota, while in MA isolated were distributed 51, 16, 10 and 4 respectively. To avoid this doubt, we have added this information to the manuscript

Q. How was the total phylum percentage calculated? Was a 16s rRNA high-throughput sequencing or just the average of M600 and marine agar bacteria?

A. This was based on the number of identified strains obtained in each media. Thus, of the 158 identified isolates, 96 total Actinomycetota were identified, and the percentage is 96/158 or 60%.

Q. To avoid confusion, please keep the taxonomic names to be consistent among figures and texts. Such as Pseudomonadota was used in the text, but Proteobacteria was used in figures.

A. By lapse, the old taxonomic names were kept. We thank the reviewer and as suggested, figures have been amended.

Q. In line 243, Fig. 3 doesn’t show the data claimed in the text.

A. This has been fixed to point to the correct figure (Fig S1)

Q. Lots of culture media were used in this manuscript, for the convenience of follow-up research from other labs, please provide the gradients of media, or catalogs if possible.

A. The full culture media formulas have been added to the text

…medium M600 [0.1% w/v peptone, 0.1% w/v yeast extract, 5 mM Tris-HCl pH 7.5, 0.1% w/v glucose, 0.1% v/v of Vitamin solution (0.1 μg mL−1 Cyanocobalamin, 2.0 μg mL−1 biotin, 5.0 μg mL−1 thiamine-HCl, 5.0 μg mL−1 Ca-pantothenate, 2.0 μg mL−1 folic acid, 5.0 μg mL−1 riboflavin, 5.0 μg mL−1 nicotinamide) and 0.2% v/v of Hutners solution (3 99 mg/L FeSO4.7H2O, 12.67 mg/L NaMoO4.2H2O, 3.34 g/L CaCl2.2H2O, 29.70 g/L MgSO4.7H2O, 50 mL/L “44” Metals solution, and 10.0 g/L Nitrilotriacetic acid. For 100 mL of “44” Metals: 250 mg Ethylenediaminetetraacetic acid, 1095 mg ZnSO4.7H2O, 500 mg FeSO4.7H2O; 154 mg MnSO4.H2O, 39.2 mg CuSO4.5H2O; 24.8 mg Co(NO3)2.6H2O, 17.7 mg Na2B4O7.10H2O] [32] and Marine agar (MA) (0.5% w/v peptone, 0.1% w/v yeast extract, 1L aged natural seawater)…

Q. Please add a reference for OSMAC.

A. The following reference has been added to the phrase

“The one strain many compounds (OSMAC) [45] approach was performed with the 12 bioactive strains in more or less oligotrophic media, 1:10 M607…”

Romano S, Jackson SA, Patry S, Dobson ADW. Extending the "One Strain Many Compounds" (OSMAC) Principle to Marine Microorganisms. Mar Drugs. 2018 Jul 23;16(7):244. doi: 10.3390/md16070244. PMID: 30041461; PMCID: PMC6070831.

Q. Though iChip is talented in primary enrichment of marine bacteria, the M600 or marine agar-based culture definitely would result in biased enrichment of specific types of bacteria, which may explain the massive difference in microbiota composition from different researches. Meta-genomic sequencing-based research after iChip enrichment may provide a more in-depth analysis of new bactericidal natural products and their producers.

A. While this is particularly interesting in a study like this, we currently do not have samples to perform this.

Reviewer 3 Report

The manuscript «iChip inspired isolation, bioactivities and dereplication of Actinomycetota from Portuguese beach sediments» matches the topic of a Special Issue. In this study, using the iChip technique, 158 isolates were obtained from marine sediments of Memoria beach, Portugal. The authors have done a great job at identification and analysis of the isolates. To study the antibiotic activity, the authors used the one strain many compounds (OSMAC) approach. In situ cultivation approaches are of high interest for natural products studies and the manuscript is an excellent example of the vast biotechnological potential of an untapped marine biodiversity. The manuscript is certainly of high interest to readers of Microorganisms.

There are few suggestions that might improve the presentation of the results:

11.  A schematic diagram of iChip design, including more details, e.g. the exact material of the membrane, might be of use in the Materials and Methods section

22. If the cultivation device was sealed on the top, how was aeration and moisturizing in the wells achieved?

33. Figures 2,3: The figure would benefit from colors and bigger font size.

  4. Do the negative values of growth inhibition in Table 2 imply increased growth of the test culture?

Author Response

Answers for Comments and Suggestions of Reviewer 3

Q. A schematic diagram of iChip design, including more details, e.g. the exact material of the membrane, might be of use in the Materials and Methods section

A. Figure 1 was amended with this advice in mind. We hope it is in line with the intended goal of the reviewer.

Q. If the cultivation device was sealed on the top, how was aeration and moisturizing in the wells achieved?

A. As shown in the new Figure 1, plate wells can communicate with the wet sediment thought a 0.22 um filter. We also amended the text to clarify this in the manuscript.

Q. Figures 2,3: The figure would benefit from colors and bigger font size.

A. Figures were changed accordingly

Q. Do the negative values of growth inhibition in Table 2 imply increased growth of the test culture?

A. Yes, in those cases, compared with the control, the target had grown more and thus a negative percentage was obtained.